

# The relationships between toad behaviour, antipredator defences, and spatial and sexual variation in predation pressure

Francisco Javier Zamora-Camacho[1,2]

[1] Departamento de Sistemas Físicos, Químicos y Naturales, Universidad Pablo de Olavide, Seville, Spain
[2] Departamento de Biodiversidad, Ecología y Evolución, Universidad Complutense de Madrid, Madrid, Spain

## ABSTRACT

**Background:** Animal behaviour is under strong selection. Selection on behaviour, however, might not act in isolation from other fitness-related traits. Since predators represent outstanding selective forces, animal behaviour could covary with antipredator defences, such that individuals better suited against predators could afford facing the costs of riskier behaviours. Moreover, not all individuals undergo equivalent degrees of predation pressure, which can vary across sexes or habitats. Individuals under lower predation pressure might also exhibit riskier behaviours.
**Methods:** In this work, I tested these hypotheses on natterjack toads (*Epidalea calamita*). Specifically, I gauged activity time, exploratory behaviour and boldness in standard laboratory conditions, and assessed whether they correlated with body size and antipredator strategies, namely sprint speed, parotoid gland area and parotoid gland colour contrast. Additionally, I compared these traits between sexes and individuals from an agrosystem and pine grove, since there is evidence that males and agrosystem individuals are subjected to greater predation pressure.
**Results:** Sprint speed as well as parotoid gland contrast and size appeared unrelated to the behavioural traits studied. In turn, body mass was negatively related to activity time, boldness and exploration. This trend is consistent with the fact that larger toads could be more detectable to their predators, which are mostly gape unconstrained and could easily consume them. As predicted, females exhibited riskier behaviours. Nonetheless, agrosystem toads did not differ from pine grove toads in the behavioural traits measured, despite being under stronger predation pressure.

# INTRODUCTION

Animal behaviour is strongly subjected to selection, and thus represents a fundamental component of fitness (*Dingemanse & Réale, 2005*; *Dugatkin, 2020*). Traits such as sexual selection (*Schuett, Tregenza & Dall, 2010*), reproductive success (*Zhao et al., 2016*), productivity (*Biro & Stamps, 2008*), contest outcome (*Briffa, Sneddon & Wilson, 2015*), and even mortality (*Stamps, 2007*), are known to be linked to traits such as boldness (*i.e.,* willingness to engage in activities that involve exposure) or exploratory behaviour (*i.e.,*

Corresponding author
Francisco Javier Zamora-Camacho, zamcam@ugr.es

willingness to investigate a novel environment), although these traits oftentimes exert opposing fitness consequences. Indeed, selection on behaviour can hardly be regarded as directional, since the fitness consequences of behavioural traits are context-dependent (*Smith & Blumstein, 2008*; *MacPherson et al., 2017*). For instance, bold red squirrels (*Sciurus vulgaris*) survive better in food-restrictive habitats, but worse in sites where food supplies are stable, whereas exploratory behaviour has a consistent negative relationship with survivorship and female reproductive success across habitats (*Santicchia et al., 2018*). This case illustrates how the diversity of contexts animals can face could be key in maintaining the enormous variation in behavioural traits documented (*Nettle, 2006*; *Briffa & Sneddon, 2016*; *Roche, Careau & Binning, 2016*). Nevertheless, the contextual components of animal behavioural traits are poorly understood and remain an eminent subject of debate among scientists (*Koski, 2014*; *Weiss, 2018*; *Wilson et al., 2019*).

Particularly relevant are the fitness consequences of behavioural traits that affect mortality. With predators being among the most frequent causes of mortality in many animals (*Lima, 2002*; *Beauchamp, Wahl & Johnson, 2007*), the substantial effect of antipredator behaviour on prey fitness comes as no surprise (*Lind & Cresswell, 2005*). Prey's behavioural traits are tuned to, and affected by, predation pressure in intricate ways (*Toscano & Griffen, 2014*; *Belgrad & Griffen, 2016*). On the one hand, predators may trigger plastic behavioural responses in their prey (*Quinn & Cresswell, 2005*; *Dingemanse et al., 2010*). For instance, *Euricea nana* salamanders reduce their activity time in the presence of predators (*Davis & Gabor, 2015*). Likewise, *Parus major* tits exposed to predators have a lower tendency to explore than non-exposed controls (*Abbey-Lee, Mathot & Dingemanse, 2016*), and juvenile *Negaprion brevirostris* sharks that exhibit more exploratory behaviour forage in riskier habitats, but only under low predation pressure (*Dhellemmes et al., 2021*). On the other hand, the relationships between predation risk and behavioural traits may transcend plasticity, as behaviour can be persistent across environments (*Dosmann & Mateo, 2014*). In such a scenario, the ways in which behaviour affects predation risk can be complex. For example, shy *Rutilus rutilus* roaches are more likely to be preyed on by ambushing predators than bold conspecifics (*Blake et al., 2018*), whereas bolder *Panopeus herbstii* crabs experience greater mortality rates (*Belgrad & Griffen, 2018*).

Despite the fact that some trends have been detected, inter-individual variation in antipredator behavioural traits is oftentimes high (*López et al., 2005*; *Brown et al., 2014*; *Cremona et al., 2015*). Such variation could be maintained by spatial differences in predation pressure driving diverging behavioural traits. Supporting this possibility, *Phoxinus phoxinus* minnows from a population under greater predation pressure are bolder but less active than conspecifics from a population where predators are less abundant (*Kortet et al., 2015*). Indeed, how animals make use of space affects their success against predators (*Leblond, Dussault & Ouellet, 2013*). Moreover, whenever the sexes are subjected to differential predation pressures, sexual disparities in behaviour could be expected. For instance, highly active *Perca fluviatilis* perch males face greater mortality than females (*Yli-Renko, Pettay & Vesakoski, 2018*), and *Poecilia reticulata* guppy males are significantly bolder than females (*Harris et al., 2010*).

The protection lent by antipredator defences of different kinds could buffer the effects of predation on behavioural traits. Other potential sources of variation in antipredator behaviour, however, remain underexplored-such as locomotion, chemical deterrents, and colour. One of the most widespread antipredator defences is locomotion, as an active flight can be efficient in avoiding predators (*Watkins, 1996*; *McGee et al., 2009*). Although locomotion has a behavioural component, as animals tune their investment on locomotion according to the benefits it can yield in different situations (*Zamora-Camacho, García-Astilleros & Aragón, 2018*), it may also function as a capability that is dependent on traits other than behaviour (*Zamora-Camacho, 2018*). Other antipredator defences are neatly non-behavioural and passive. Such is the case with chemical deterrents, which are toxic or distasteful substances that repel predator attacks (*Mebs, 2001*; *Brodie, 2009*; *Savitzky et al., 2012*). Aposematic coloration (*i.e.*, conspicuous colours and patterns that potential predators associate with unpalatability and actively avoid) can be frequently found alongside chemical deterrents(*Saporito et al., 2007*; *Zvereva & Kozlov, 2016*; *Ruxton et al., 2018*), thus providing their carriers with additional defence from predators (*Skelhorn & Rowe, 2006*; *Prudic, Skemp & Papaj, 2007*). However, the potential link between these antipredator strategies and behavioural traits requires further exploration.

The relationship between body size and predation is particularly interesting. On the one hand, larger prey could be easier to detect (*Mänd, Tammaru & Mappes, 2007*; *Karpestam, Merilaita & Forsman, 2014*), while, on the other hand, larger prey can be more difficult to handle (*Díaz & Carrascal, 1993*; *Kalinkat et al., 2013*). In fact, survivorship of *Hyla chrysoscelis* tadpoles to attacks by *Tramea lacerata* dragonfly nymphaea increases with body size (*Semlitsch, 1990*). Larger grasshoppers are also better defended against a wide array of predators (*Whitman & Vincent, 2008*). However, larger predators do not necessarily prefer larger prey (*Tsai, Hsieh & Nakazawa, 2016*), although they can exploit prey of a wider size range (*Radloff & du Toit, 2004*). The influence of the morphology of a predator's mouthparts is also important, with gape-limited predators preferring smaller prey, while gape-unconstrained predators are less dependent on their prey's body size (*Jobe, Montaña & Schalk, 2019*). Furthermore, the role of prey body size may depend on other antipredator strategies. For example, whereas detectability of cryptic prey can appear unrelated to body size, conspicuous prey might be more detectable at larger body sizes (*Mänd, Tammaru & Mappes, 2007*). Moreover, toxins are more efficient antipredator defences in smaller prey (*Smith, Halpin & Rowe, 2016*). Consequently, antipredator behaviour is not independent from body size, but such relationships are intricate (*Preisser & Orrock, 2012*).

In this work, I studied activity, exploratory behaviour, and boldness in the natterjack toad, *Epidalea calamita*. Specifically, I tested whether these traits co-vary with other antipredator strategies, including locomotion. This toad is cursorial, and uses quick runs to flee from its predators (*Zamora-Camacho, 2018*). I also examined the potential effects of body size on these behavioural traits. Additionally, this species has notable parotoid glands, which are a pair of swollen structures located dorsally behind the eyes in many amphibian species. The size of these glands is directly proportional to the quantity of chemical deterrents they are capable of ejecting (*Zechmeister, 1948*; *Llewelyn et al., 2012*).

The parotoid glands of *E. calamita* are aposematic, with predators avoiding plasticine models with highly colour-contrasting parotoid glands (*Zamora-Camacho, 2021*). I tested whether the aforementioned traits are correlated with sprint speed and parotoid gland area and colour contrast. In addition, predation pressure is subject to spatial variation, according to an experiment where plasticine toad models received more attacks in an agrosystem than in a natural habitat (*Zamora-Camacho, 2021*). Agrosystem toads are larger than those from a natural habitat (*Zamora-Camacho & Comas, 2017*), which could be an adaptive response to greater predator pressure. Accordingly, I tested whether the aforementioned traits vary between these habitats. Finally, the fact that males in this species are faster (*Zamora-Camacho, 2018*) and have larger parotoid glands than females (*Zamora-Camacho, 2021*) supports the notion that males are under greater predation pressure than females, which aligns with reports from other related species (*e.g.*, *Frétey et al., 2004*). I tested whether the aforementioned traits vary between sexes. I predict that individuals that are better suited against predators (*i.e.*, those that are faster, larger in body size, or have larger and more contrasting parotoid glands) will display riskier behavioural traits, including being more active, bolder, and more exploratory. Similarly, I expect pine grove toads and females to be more active, bolder, and more exploratory, as they are under reduced predation pressure.

## MATERIALS AND METHODS

### Study species

*Epidalea calamita* is a bufonid toad that thrives in diverse habitats, including unaltered as well as human-modified systems, in extensive areas in central and western Europe (*Gomez-Mestre, 2014*). Owing to the variable climatic conditions throughout this vast area, the phenology of this species is asynchronous, with aestivation being common in hot regions and hibernation occurring in cold climates (*Gomez-Mestre, 2014*). This species is primarily nocturnal, and its activity and reproduction take place during wet and not excessively cold weather, which happens during winters in warmer regions and in the spring in colder regions (*Gomez-Mestre, 2014*). Under adverse circumstances, they rest under rocks or logs, or in dens they burrow in loose soils, safe from predators (*Gomez-Mestre, 2014*). These toads are potential prey of a wide array of predators, including snakes (*e.g.*, *Natrix maura* and *Natrix astreptophora*), birds (*e.g., Larus ridibundus* and *Pica pica*) and mammals (*e.g., Meles meles*), among others (see *Gomez-Mestre, 2014*). When under attack, toads use intermittent runs to flee (*Zamora-Camacho, 2018*). When escape is not possible, however, they commonly arch their loins and exhibit their parotoid glands, which can release great amounts of toxins (*Stawikowski & Lüddecke, 2019*).

### Animal capture and management

Toads were captured in the pine grove "Pinares de Cartaya" (SW Spain: 37°20′N, 7°09′W) and in the agrosystem nearby. The forest is an 11,000-ha extension dominated by *Pinus pinea* and an undergrowth of Mediterranean bushes such as *Pistacea lentiscus*, *Cistus ladanifer* and *Rosmarinus officinalis*. Although this plant assemblage could be considered autochthonous or introduced in this region, its predominance dates back at least

4,000 years (*Martínez & Montero, 2004*), and as such it is deemed a natural habitat for the purposes of this study. The agrosystem is about 5 km away from the pine grove, and is a 2,800-ha agricultural area where extensive vegetable crops have gradually given way to intensive orange tree, blueberry, and strawberry fields (among others) throughout the last few decades. In these croplands, landowners apply fertilizers, fungicides, herbicides, and pesticides at their discretion, and artificial watering softens the three-to-four-month-long summer droughts. Animal capture and management was according to permits by the Junta de Andalucía government (Reference AWG/mgd GB-369-20).

Due to the mild local climate, *E. calamita* breeds in the winter there. Accordingly, toad capture was conducted from December 2018 to March 2019. I caught 22 females and 20 males in the agrosystem, plus 21 females and 25 males in the pine grove. Toads were captured by hand while active in nights of suitable weather, then transported to the laboratory in plastic buckets with well-ventilated lids and a substrate of humid earth. When they were in the laboratory, I used their sexual dimorphism in coloration (females have browner backs and greyish throats, whereas males have greener backs and purplish or pinkish throats; *Zamora-Camacho & Comas, 2019*) and the presence of blackish nuptial pads on male forelimbs (*Gomez-Mestre, 2014*), to sex them. Next, I allocated them to individual plastic terraria (20 × 13 × 9 cm) with wet peat as a substrate and an opaque plastic shelter. Toads were undisturbed in these terraria at all times, except during the trials. Photos were taken approximately 24 h after capture (see below). Then, 24 h after the photos, toad activity trials were recorded (see below). Finally, 24 h after the activity trials, sprint speed tests were performed (see below). Toads were released at their capture sites shortly afterward.

## Measurements of coloration and morphology

I used a ruler to measure toad snout-vent length (hereafter, SVL) to the nearest mm, and a scale (model CDS-100) to weigh them to the nearest 0.01 g. No later than 24 h after capture, I orthogonally photographed each toad's back using a photo camera Canon EOS 550D, set at 18 megapixels of resolution, F10 of shutter-aperture, and a focal length fixed at 53 mm. Only exposure time was allowed to be automatically adjusted by the device, to optimize sharpness in each individual photo. The camera was secured to a tripod, which guaranteed perpendicularity, steadiness, and a constant distance of 40 cm from the lens to the photographed area. This area was a square (30 cm side), white piece of paper that lay horizontal. On both lateral and the rear sides (considering that the tripod was located opposite to the front side), three square (30 cm side), white pieces of white polyester sat vertically, conforming an incomplete cube which lacked the front (allowing toad handling) and the upper sides (allowing photograph taking). In order to avoid all parasitic lights (*i.e.*, any uncontrolled source of light), photos were taken at night in a completely closed room, where the only sources of light were two 80W white-light bulbs, one next to each lateral side of the cube, externally to it, at a height of 20 cm, so that shades on the photographed area were prevented and the white polyester of the lateral, vertical squares filtered the light. This setting is depicted in Fig. S1. Immediately prior to taking the photos, once the set was fixed as described, I calibrated white balance to a spotless piece of

paper, after which I added a standardized colour chart (IMAGE Photographic) for digital calibration of white balance, and a piece of graph paper to calibrate length. Any remainder of humidity and dirt was gently removed from the toads' skins with a disposable napkin before each photo.

Afterwards, these photos were processed with the software Adobe Photoshop CS5. Firstly, I calibrated white balance one more time in each photo by using the tool eyedropper in the white calibration function on the colour chart. Furthermore, colour mode was set to the L*a*b* colour space preconized by the *Commision Internationale d'Eclairage* (CIE) (*Montgomerie, 2006*). This is a three-dimensional colour space were L* quantifies lightness, and varies from 0 (pure black) to 100 (pure white); a* quantifies the green-red axis (positive values represent red and negative values represent green); and b* quantifies the blue-yellow axis (positive values represent yellow and negative values represent blue). I calibrated length using the piece of graph paper, and manually outlined both parotoid glands making use of the lasso tool, which allowed me to calculate the sum of the areas of each. After this, parotoid gland relative area was calculated as the residuals of regression of parotoid gland area against SVL. Once the parotoid glands were outlined, I calculated their average colour and, with the histogram tool, retrieved their average values of L*, a*, and b*. Lastly, I followed the same steps to trace the dorsum (excluding the parotoid glands and the limbs) and retrieve its average L*, a*, and b* values. The average L*, a*, and b* parotoid gland and dorsum values were used to calculate parotoid gland contrast ($\Delta E^*$) as in the CIE formula to assess difference in colour: $\Delta E^* = (\Delta L^{*2} + \Delta a^{*2} + \Delta b^{*2})^{1/2}$ (*Nguyen, Nol & Abraham, 2007*; *Moreno-Rueda et al., 2019*).

## Measurements of activity and sprint speed

Starting 24 h after the photos were taken, toads were recorded for activity and sprint speed trials (see details below), in this order. Videos were filmed with a camera Canon EOS 550D, at 25 frames per second. The camera was attached to a 2.5 m high tripod, with a 90° angle, at all times. In both trials, only one individual was recorded at a time. To remove the effect that temperature may have on amphibian activity (*Muller, Cade & Schwarzkopf, 2018*) and locomotion (*Preest & Pough, 2003*), the room was at approximately 19 °C at all times. Light conditions were standardized, as the only light source during all trials was a 60 W white light bulb 2.5 m high at the centre of the container where the toad was performing the trial in question (see below). All videos were recorded at night (approximately between 21:00 and 02:00, local time), when these toads are naturally active (*Gomez-Mestre, 2014*).

## Measurement of activity

Activity trials were recorded while these toads were freely moving in a plastic arena (54 × 27 × 40 cm). A grid of 9 cm side squares was painted with non-toxic ink on the bottom of this arena. Prior to the recordings, toads were placed at the centre of the arena, enclosed within a vertical hollow cylinder (50 cm high, 15 cm diameter) open at its lower end. The cylinder was built with a metal mesh (5 mm light), which allowed acclimation to the experimental setting. After 2 min, the cylinder was gently removed in

the vertical plane, and the toad's activity was recorded for 10 min (*Chajma, Kopecký & Vojar, 2020*).

Videos were then analysed with the program Tracker v. 4.92. I measured several variables as surrogates of different traits of alleged relevance in the behaviour of animals in general (*Réale et al., 2007*) and of amphibians in particular (*Kelleher, Silla & Byrne, 2018*). Activity time was the amount of time (s) the toad spent moving (*Chajma, Kopecký & Vojar, 2020*). Exploration behaviour was estimated as the number of squares visited (excluding squares that had been visited before; *Chajma, Kopecký & Vojar, 2020*) and the number of square visits (counting the number of times any square was visited, including repeated visits; *Carlson & Langkilde, 2013*). These measures differ in the fact that the former assumes that the individual distinguishes and keeps track of the areas that have already been visited, whereas the latter assumes the opposite (*Carlson & Langkilde, 2013*). Time until the first move (*i.e.*, latency) was also recorded, as a surrogate of the shyness/boldness gradient, as bolder individuals are expected to start moving sooner (*Chajma, Kopecký & Vojar, 2020*). The use of space is also widely considered a surrogate of the shyness/boldness gradient, in amphibians (*Réale et al., 2007*; *Carlson & Langkilde, 2013*; *Chajma, Kopecký & Vojar, 2020*) and other taxa (*Burns, 2008*; *Harris, D'Eath & Healy, 2009*). Specifically, thigmotaxis, the tendency for some individuals to remain in the periphery of their enclosures next to the walls rather than in the open areas, has been regarded as an anxiety-like, predator-avoidance behaviour as opposed to the boldness subjacent to the use of open areas (*Harris, D'Eath & Healy, 2009*; *Carlson & Langkilde, 2013*; *Chajma, Kopecký & Vojar, 2020*). Therefore, I also estimated the shyness/boldness gradient by counting independently the number of external and internal squares visited (excluding squares that had been visited before) and the number of external and internal square visits (counting the number of times squares of these types were visited, including repeated visits). Then, I divided the number of external squares visited by the total number of squares visited (external squares visited ratio), as well as the number of external square visits by the total number of square visits (external square visit ratio). Both ratios have a direct relationship with boldness. These measurements are relevant in an ecological context, as laboratory surrogates of animal behaviour mirror actual behaviour in the wild (*Herborn et al., 2010*).

## Measurement of sprint speed

Prior to conducting and recording the sprint speed trials, I emptied toad bladders by firmly-but gently-pressing their lower abdomens, which eliminates any potential effect of different bladder water burden by reducing it to zero (*Preest & Pough, 1989*; *Walvoord, 2003*; *Prates et al., 2013*). Next, I allowed toads to rest in their terraria for 1 h. After that, they were recorded (with the same camera already described) while running along a brownish cardboard linear raceway (200 × 15 × 15 cm). On its bottom, one transversal white stripe of insulating tape was placed every 10 cm, so that the raceway was longitudinally divided into 10-cm stretches delimited by contrasting-colour stripes easy to visualize in the videos. Locomotor performance may depend on the substrate where the race takes place (*Vanhooydonck et al., 2015*): cardboard provided a surface rough enough
to facilitate an appropriate traction. I also set a dark background at one end of the raceway, which could be viewed as a shelter and encourage toads' moving forward (*Zamora-Camacho, 2018*; *González-Morales et al., 2021*). Individuals were placed at the opposite end of the raceway, and continuously pursued as a way of encouraging running, until they covered the raceway. Once these trials were completed, toads were released at their capture sites within 24 h. No visible damage was inflicted on toads because of this investigation.

The footages produced were analysed with the program Tracker v. 4.92, which allows frame-by-frame video handling. For each toad, I registered the time (to the nearest 0.01 s) needed to cover each stretch in the raceway, which equals the time elapsed between the moments when the snout of a toad surpassed two consecutive white stripes (*Martín & López, 2001*; *Zamora-Camacho et al., 2014*; *Zamora-Camacho, 2018*). As the distance covered was 10 cm in all cases, I calculated the speed (in cm/s) reached in each stretch by dividing 10 cm by the time (s) it took for the toad to cover it. I considered each individual's sprint speed as its highest speed value. Finally, relative speed was calculated as the residuals of regression of sprint speed against SVL.

### Statistics

Firstly, I built two correlation matrices, one including the behavioural traits measured (number of squares visited, number of square visits, external squares visited ratio, external square visits ratio, activity time, and time until the first move) and another including the antipredator defences measured (body mass, parotoid gland contrast, parotoid gland relative area, and relative sprint speed). The aim of these matrices was to detect collinearity between both sets of data. Most behavioural traits measured were highly correlated, except for time until the first move (Table S1). On the contrary, the antipredator defences measured were mostly uncorrelated, except for relative sprint speed and parotoid gland relative area, which were positively and significantly correlated (Table S2).

Then, to condense the correlated variables into fewer, uncorrelated variables, and solve the limitation caused by the high collinearity among the variables measured, I conducted a Principal Component Analysis (PCA; *Jongman, Braak & Tongeren, 1995*) including the behavioural traits measured that were correlated (number of squares visited, number of square visits, external squares visited ratio, external square visits ratio, and activity time; Table S3a) and another PCA including the antipredator defences measured that were correlated (parotoid gland relative area and relative sprint speed; Table S3b). In both cases, only Principal Components (PC) with an eigenvalue greater than 1 were selected, according to the Guttmann-Kaiser Criterion (*Yeomans & Golder, 1982*).

Then, I conducted three separate ANCOVAs, where habitat, sex and their interaction were included as factors, and all PC with an eigenvalue greater than 1 in the second PCA (namely, PC1, see Results below), as well as body mass and parotoid gland contrast, were included as covariates. The response variable of the first ANCOVA was time until the first move. In the second and the third ANCOVAs, the variable responses were each PC with an eigenvalue greater than 1 in the first PCA (namely, PCa and PCb, see Results

below). Stepwise backward selection was applied to these ANCOVAs. Tests were conducted with the package "*nlme*" (*Pinheiro et al., 2012*) in the software R (*R Development Core Team, 2012*). Before conducting parametric statistical analyses, I checked that the data met the criteria of homoscedasticity and residual normality (*Quinn & Keough, 2002*). Since no transformation could make body mass homoscedastic, I implemented the function "*varIdent*" (*Zuur et al., 2009*).

## RESULTS

### Variable grouping according to PCAs

In the PCA including the behavioural traits, the only two PC with eigenvalues greater than 1, named PCa and PCb, explained jointly 88.99% of the total variance. PCa was strongly and positively correlated with the number of squares visited, the number of square visits, and activity time, whereas its correlations with the external squares visited ratio and the external square visits ratio were strong and negative (Table 1). Therefore, PCa was positively correlated with exploration behaviour, activity, and boldness. In turn, PCb was negatively correlated with all behavioural traits measured (Table 1). Therefore, PCb was negatively correlated with exploration behaviour and activity, and positively correlated with boldness.

In the PCA including parotoid gland relative area and relative sprint speed, the only PC with an eigenvalue greater than 1, named PC1, explained 68.20% of the total variance. PC1 was strongly and positively correlated with relative sprint speed and relative parotoid gland area (Table 2).

### ANCOVAs

After stepwise backward selection was applied to the ANCOVA including PCa as the response variable, sex had a significant effect, with PCa being greater in females than in males (Mean ± SE; females: 0.387 ± 0.220; males: −0.349 ± 0.300; $X^2_{1, 85} = 3.910$; $P = 0.048$; Fig. 1). Moreover, body mass had a negative, significant relationship with PCa ($X^2_{1, 85} = 8.122$; $\beta = -0.045$; $P = 0.004$).

After stepwise backward selection was applied to the two ANCOVAs including either PCb or time until the first move as response variables, no factor or covariate appeared significant in either case.

## DISCUSSION

My findings show that the antipredator defences of *E. calamita* are linked to behaviour in some cases, but not in others. For example, relative speed as well as parotoid gland contrast and size were not related to the traits studied. In contrast with other bufonids, such as *Rhinella marina*, whose jumping distance is directly proportional to body length (*Hudson et al., 2020*), sprint speed of *E. calamita* is unrelated to body size (*Zamora-Camacho, 2018*), which allows for a separate evaluation of both parameters. The absence of correlations between speed and the behavioural traits tested as a part of this study is aligned with findings on *Zootoca vivipara* (*Le Galliard et al., 2013*) and *Phrinocephalus vlangalii* lizards (*Chen et al., 2019*), but not with findings on *Myotomis unisulcatus* (*Agnani et al., 2020*) and

**Table 1 Correlations of Principal Components with eigenvalues greater than 1 (PCa and PCb) with each correlated behavioural variable measured.**

| Variable | PCa | PCb |
|---|---|---|
| Number of squares visited | 0.888 | −0.292 |
| Number of square visits | 0.850 | −0.412 |
| External squares visited ratio | −0.790 | −0.578 |
| External square visits ratio | −0.766 | −0.610 |
| Activity time | 0.774 | −0.406 |

Note:
PCa was strongly correlated with all behavioural variables included in the PCA, this correlation being positive with number of squares visited, number of square visits, and activity time, and negative with external squares visited ratio and external square visits ratio. PCb was negatively correlated with all behavioural variables included in the PCA, with weak or medium correlations.

*Tamias striatus* rodents (*Newar & Careau, 2018*). It is worth mentioning that all toads in this sample engaged in flight behaviour. However, the tonic immobility observed in other cases, especially in taxa that rely on their toxins against attacks, can also be considered an expression of boldness (*Edelaar et al., 2012*; *Hudson, Brown & Shine, 2017*). In turn, relationships between overall coloration and different behavioural traits have been described in taxa as disparate as tortoises (*Mafli, Wakamatsu & Roulin, 2011*), fish (*Schweitzer, Motreuil & Dechaume-Moncharmont, 2015*) and birds (*Costanzo et al., 2018*). However, there is a lack of such studies on amphibians (reviewed in *Kelleher, Silla & Byrne, 2018*) and, to the best of my knowledge, the potential relationship between aposematism and behaviour at the individual level remains unexplored thus far. According to these results, the degree of aposematism is unrelated to behaviour in these toads. A decoupling between colour and behaviour, albeit in a reproductive context, has also been described in phrysonomatid lizards (*Wiens, 2000*). These mismatches between some antipredator defences and behaviour could suggest that the success of these traits is independent of each other, or simply that they have evolved separately.

Similarly, parotoid gland size was not associated with a more exploratory and bolder behaviour nor increased activity time. These are considered risky behavioural traits that increase conspicuousness to predators (*Hall et al., 2015*; *Reader, 2015*). This finding does not support the prediction that more extensive parotoid glands, capable of releasing greater amounts of toxins (*Zechmeister, 1948*; *Llewelyn et al., 2012*), could better protect their bearers against predators, thus reducing the potential costs of risky behaviours with regard to their benefits (*Smith & Blumstein, 2008*; *Niemelä, Lattenkamp & Dingemanse, 2015*). However, the amount of toxin contained in the glands at the moment of the trials could not be assessed. This could represent a limitation of the experimental design, as closely-related *R. marina* toads adjust their antipredator behaviour after parotoid gland toxin depletion (*Blennerhassett et al., 2019*). On the contrary, body mass was negatively related to exploratory behaviour, boldness, and activity time. Although larger prey individuals could be better suited against gape-limited predators (*Turesson, Persson & Brönmark, 2002*; *Urban, 2007*), this is not necessarily true when predators are non-gape-limited (*Jobe, Montaña & Schalk, 2019*; *Stretz, Andersson & Burkhart, 2019*). Indeed, gape

**Table 2 Correlations of the Principal Component with an eigenvalue greater than 1 (PC1) with each correlated antipredator defence measured.**

| Variable | PC1 |
| --- | --- |
| Parotoid gland relative area | 0.826 |
| Relative sprint speed | 0.826 |

Note:
PC1 was strongly and positively correlated with both antipredator defences included in the PCA, namely parotoid gland relative area and relative sprint speed.

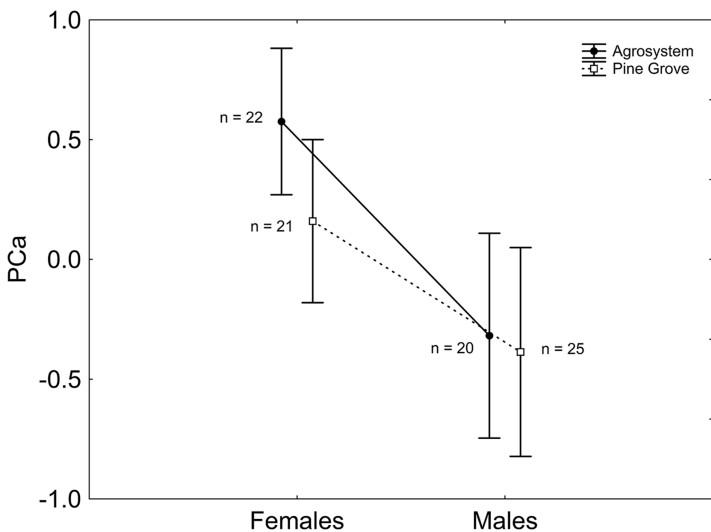

**Figure 1 Sex and habitat differences in PCa.** PCa was greater in females than in males, but did not differ between habitats. Note that PCa was positively correlated with the number of squares visited, the number of square visits, and activity time, and negatively correlated with the external squares visited ratio and the external square visits ratio. Vertical bars represent standard errors. Sample sizes are indicated.

unconstrained predators such as mammals (*Owen-Smith & Mills, 2008*) or birds (*Comay & Dayan, 2018*) can and do handle remarkably voluminous prey. Although some local snakes (mainly *Natrix astreptophora* and *N. maura*; *Gomez-Mestre, 2014*), which are gape-limited, have been described as predators of these toads, their activity seldom overlap, as those snakes are mainly diurnal and hibernate in the winter (*Santos, 2015*; *Pleguezuelos, 2018*), whereas *E. calamita* toads are primarily active in winter nights (*Gomez-Mestre, 2014*). Therefore, the most likely predators of *E. calamita* adults are birds and mammals (*Gomez-Mestre, 2014*), to which larger toads could be more conspicuous, but not less vulnerable. In this context, the less risky behaviour of larger toads could be advantageous against their main predators. Variation in the relative pressure exerted by dissimilar predator preferences on body size might underlie the apparently contradictory relationships between body size and behaviour found among this and other studies. For example, whereas bold *Lacerta monticola* male lizards are smaller (*López et al., 2005*), there is a positive relationship between body size and boldness in juvenile *Tropidonophis mairii* snakes (*Mayer, Shine & Brown, 2016*) and between body size and exploratory behaviour in *Pseudophryne corroboree* frogs (*Kelleher et al., 2017*).

Moreover, females exhibited a bolder behaviour than males. This finding is aligned with the assumption that female toads are under milder predation pressure (*Frétey et al., 2004*), and can thus afford riskier behaviours. *E. calamita* males in this system are faster (*Zamora-Camacho, 2018*), brighter (*Zamora-Camacho & Comas, 2019*), and have larger parotoid glands than females (*Zamora-Camacho, 2021*), which could be interpreted as antipredator defences triggered by a harsher predation pressure. Remarkably, other toads, such as *R. marina* (*Gruber et al., 2018*) or *Sclerophrys gutturalis* (*Baxter-Gilbert, Riley & Measey, 2021*) do not appear to differ in these behavioural traits. Sexual differences in boldness, moreover, vary notably in other taxa: male dogs (*Starling et al., 2013*) and *Brachyraphis episcopi* fish (*Brown, Jones & Braithwaite, 2007*) are bolder than females, female *Diomeda exulans* albatross are bolder than males (*Patrick, Charmantier & Weimerskirch, 2013*), and *Diploptera punctata* male and female cockroaches do not diverge in boldness (*Stanley, Mettke-Hofmann & Preziosi, 2017*).

Lastly, habitat did not affect the behavioural traits measured, despite the fact that this species is under greater predation pressure in agrosystem than in pine grove (*Zamora-Camacho, 2021*). Other traits seem to be aligned with this spatial pattern of predation pressure: agrosystem toads have a more intermittent locomotion mode (*Zamora-Camacho, 2018*), are brighter (*Zamora-Camacho & Comas, 2019*), and have larger and more contrasting parotoid glands than pine grove conspecifics (*Zamora-Camacho, 2021*), which could signify greater antipredator defences, likely triggered by more intense predation pressure. Remarkably, while habitat alone does not innately affect boldness behaviour of *S. gutturalis* tadpoles (*Mühlenhaupt et al., 2022*), predation pressure can explain spatial divergence in behavioural and morphological traits of other anurans, such as *Bombina variegata* toads (*Kang et al., 2017*). Spatial differences in boldness may have implications at other levels.

## CONCLUSIONS

To conclude, relative speed as well as parotoid gland contrast and size appeared unrelated to the behavioural traits studied. In turn, body mass was negatively related to activity time, boldness and exploration. This trend is consistent with the fact that most predators of this species are gape unconstrained and could more easily find and hunt larger toads. Females were bolder, which matches the assumptions that males and agrosystem toads are under harsher predation pressure. Nonetheless, the behavioural traits measured did not vary between habitats, which is not aligned with previous findings that agrosystem toads are under greater predation pressure. Jointly, these results partly support the predictions that behaviour is tuned to antipredator defences and to differential predation pressure in this toad. In the light of these results, disentangling the potential links between behaviour and antipredator defences, so far underexplored, could be key in the understanding of predator avoidance.

## ACKNOWLEDGEMENTS

Gregorio Moreno-Rueda and Mar Comas kindly provided their logistic support. Comments by James Baxter-Gilbert, Max Mühlenhaupt, Cameron Hudson and an anonymous reviewer improved the manuscript.

### Funding

FJZ-C was partly supported by a "Juan de la Cierva-Incorporación" postdoctoral fellowship by the Spanish "Ministerio de Ciencia e Innovación". The funders had no role in study design, data collection and analysis, decision to publish, or preparation of the manuscript.

### Grant Disclosures

The following grant information was disclosed by the authors:
Juan de la Cierva-Incorporación.
Ministerio de Ciencia e Innovación.

### Competing Interests

The author declares that he has no competing interests.

### Author Contributions

- Francisco Javier Zamora-Camacho conceived and designed the experiments, performed the experiments, analyzed the data, prepared figures and/or tables, authored or reviewed drafts of the paper, and approved the final draft.

### Animal Ethics

The following information was supplied relating to ethical approvals (*i.e.*, approving body and any reference numbers):

Approval for this research was provided by the Junta de Andalucía government.

### Data Availability

The data are available in the Supplemental File.

### Supplemental Information

Supplemental information for this article can be found online at http://dx.doi.org/10.7717/peerj.12985#supplemental-information.

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
