# Peer review of "The relationships between toad behaviour, antipredator defences, and spatial and sexual variation in predation pressure"

_PeerJ, doi:10.7717/peerj.12985_

## Round 0.1 · original submission · Minor Revisions

After receiving three thoughtful and comprehensive reviews, I am pleased to see that they all agree that this is a well conceived, conducted, and presented study - a sentiment I wholly agree with. That being said, and as with all research, there are a number of improvements that have been recommended to increase the clarity, readability, and defensibility of this manuscript and study.

Notably, it appear that there is a need for some more cited background on toads behaviour, performance capacity, and antipredator (including with respect to parotid glands). In the last few years there have been a number of different toad research published examining these topics, and it would be pertinent to reference them here to better contextualise your work.

There is a need to increase the clarity in a number of areas across the paper, this is to better explain the research, study design, and rationale for methodological decisions. In doing so we can provide a clearer understanding, not only for experts reading your work to contextualise their own (the specialist audience), but also for readers that might not be familiar with these exact types of study (an interested general audience). For example, increasing the clarity and explanation for the statistical methods used (see comments from Reviewer 3), would be beneficial. Similarly, it is important to define and contextual terms like "boldness" and "exploration" so that it is clear they are distinct behavioural traits and ensure that their use in the manuscript does not lead the read to assume that they are synonymous.

Also, one factor that particularly concerns me regarding the performance metrics for sprint speed using a single trail methodology and short track size. There is quite a bit of research suggesting single trail performance metric have a hard time producing reliable 'maximal' values, and with organism that have a saltatory (hopping/bounding) locomotion longer tracks and multiple measures are often preferred - as a few "less-than-maximal" leaps can compromise the data generated. I think that you certainly need to consider this. I understand it is not possible to go back and collect more data - but then you need to properly address this issue in the manuscript and provide a valid justification for why this limited approach can be considered sound.

I have suggested that this review outcome be considered as minor revisions - as I do not think you need to rework to statistics or overhaul the writing or anything major like that - but I do want you to pay careful attention to all of the excellent comments provided by the reviewers and address these improvements thoroughly. In doing so, I am positive you will be able to only further improve how you present this solid study and bolster your ability to continue to polish this well structured manuscript.

I am looking forward to reading the next version!

·

Basic reporting

This work represents a clearly and in professionally written manuscript. The author examined how morphological, chemical, behavioral, and performance anti-predator traits are correlated and differ spatially and between sexes in a toad species. They found that body mass was negatively related to the behavioral traits examined, whereas parotoid gland area and sprint speed were not related to those behaviors. Furthermore, females showed riskier behavior than males but there was no difference in behavior between the “natural” and agrosystem habitat. Larger toads are more conspicuous to predators that are mostly gape-limited in this system, which might explain the correlation between body mass and behavior. Further, males might be more subjected to predation than females which might explain why females exhibited riskier behavior.
The author clearly is well-experienced with this system given the list of publications (some of which are by and with the author) and provides a necessary background. One figure was provided with this manuscript which I suggest to expand by adding habitat as a factor (see “Additional comments”).

Experimental design

How different traits are integrated is an important question to understand how complex phenotypes evolve. Especially in amphibians that often use different modes of defenses (i.e., behavioral, chemical, morphological, …) this question is relevant but seldomly studied. The author manages to “disentangle” the behaviors activity, exploration, and boldness by applying methods previously used in other studies examining these behaviors in amphibians. However, I believe the reviews by Réale et al. (2007) and Kelleher et al. (2018) should be referenced here, and how methodologies follow or diverge from the guidelines outlined in these reviews should be mentioned in the methods.
Réale D, Reader SM, Sol D, McDougall PT, Dingemanse NJ (2007) Integrating animal temperament within ecology and evolution. Biol Rev 82:291–318. https://doi.org/10.1111/j.1469-185X.2007.00010.x
Kelleher SR, Silla AJ, Byrne PG (2018) Animal personality and behavioral syndromes in amphibians: a review of the evidence, experimental approaches, and implications for conservation. Behav Ecol Sociobiol 72:79. https://doi.org/10.1007/s00265-018-2493-7

Validity of the findings

Given the application of a Principle Component Analysis, any correlation of the behavioral traits was removed, making the analysis statistically robust. The author makes well-stated conclusions on the basis of their results. Building up on previous studies, the author argues that male toads and larger toads are under higher predation threat which transpired in more-pronounced anti-predator traits. One constraint that I see with this work is that the exact quality of predation is unknown. Different predator species apply different senses and modes to detect prey and use different tactics to consume prey, resulting in diverging selection for anti-predator traits. In this system there is only limited knowledge about spatial and sexual differences in predator composition, limiting the conclusions that can confidently be drawn. However, I understand that this was not the main question studied here. Further, the author makes sure to address this point in the discussion.

Additional comments

I have noted some minor issues with the editing of the text that I would like to state here.
Line 50-51: I suggest changing “… how the diversity of contexts animals can be faced with could be key in …” to “… how the diversity of contexts that animals can face could be key in …” for better readability.
Line 77-78: I suggest changing to “Indeed, how animals make use of space affects …”.
Line 91: “With” instead of “of”.
Line 93: Change start of sentence to “Frequently concomitant with these substances are …”.
Line 94: Change “to” with “with”.
Line 102: “Increases” instead of “Increase”
Line 103: Use for example “However, …” instead of “Even, …”.
Line 174: Remove “in”.
Line 197: “Remainder” instead of “Remain”
Line 225: Can you please state the exact time frame that these experiments occurred within? E.g., “between 10pm and 2am”.
Line 300: Do you mean “behavioral traits measured that were correlated”?
Line 302: See above.
Line 323-324: Did you forget “was” after “PCb”? Also, I suggest stating that “PCb” was positively correlated with boldness for easier readability.
Line 333: One empty line too much in front of “Χ2”.
Figure 1: I suggest adding in the factor of habitat. Even though non-significant, habitat differences in anti-predator traits have been a major part of the research question, and thus should be presented in the figure.

·

Basic reporting

This study investigates the relationship between sex, morphology, and colouration on the antipredator behaviours of natterjack toads from two contrasting habitats. The study is straightforward and well written with some minor grammatical changes needed (see general comments for my suggestions), but otherwise quite clear.

The background information on the topic is mostly sufficient, though I think it is missing some citations on amphibian (and specifically, toad) anti-predator behaviors or parotoid gland morphology. I have suggested a few relevant citations (again, see general comments), but there are more out there that could be relevant to the discussion. A discussion of whether toads are capable of self-assessment with regards to gland size / toxicity could also be helpful to the reader. Also further discussion on the sexual differences in behavior / morphology would be insightful, as the pattern appears to be different from my experience with other bufonid species (i.e. males are more dispersive and therefore bolder, females have larger toxin glands). Contrasting the findings here on E. calamita with those of other bufonids would perhaps shed light on the way that selection from different predators drives antipredator trait/behaviour evolution.

I’m happy with the way the analyses were conducted and presented at this time, so I have no changes to suggest in that regard. Some of the raw data (e.g. parotoid gland area, sprint speed, and number of squares visited during the behavioural trials) appear to be absent however, so I would request that it be added for reproducibility (either now or upon publication).

Experimental design

The experimental design is appropriate to answer the questions posed, and the data collected is sufficient to do so. The methodology is well described and could be repeated, however I would request some additional details to be more explicit (e.g. the timeline between animal capture and each experiment; again, see comments below).

From an ethical point of view, the experiment is unlikely to cause long-lasting stress for the animals, and adheres to the rules outlined by the study permits. The only issue that I can see is that the acclimation time before exploration trials (2 minutes) might be too short (I would advise longer acclimation times in future work). The toads were likely still affected by experimenter handling, and this may have influenced their behaviour during the trials.

Validity of the findings

The findings appear to be statistically sound, and well presented. They expand upon previous work in the system and link back to the original hypotheses. The study design is also well balanced between sexes and treatments. As mentioned in part 1 above, not all raw data used appears to be provided, so that is one place where I must request additional information.

Additional comments

Line 41: Change “reproduction” to reproductive

Line 44-45: Rephrase to “Indeed, selection on behaviour can hardly be regarded as directional”

Line 48-49: Rephrase to “exploratory behaviour has a consistently negative relationship”

Line 93-96: Rephrase to “Aposematic coloration (i.e. conspicuous colors and patterns that potential predators associate to unpalatability and actively avoid) can be frequently found alongside chemical deterrents (citations), thus providing their carriers with additional defense from predators (other citations).”

Line 98: Rephrase to “The relationship between body size and predation is particularly interesting.” – or something similar

Line 98 – I think this paragraph could use additional discussion of gape limited vs. unconstrained predators. This comes up later in the discussion, and is suggested to be the reason why larger toads are less bold, so I think the idea should be raised earlier in the introduction.

Line 147 – I would change “reptiles” to snakes, because the predation mode by snakes here is important for the context and both example predators listed are snakes. Unless there are also lizard predators that are relevant for E. calamita? But I am not aware of any.

Line 176 – Please include a statement of how long animals were held in captivity before the trials. I believe later on in the text it states that photographs were taken 24h post capture, and behavioural trials 24h after that? I think a clearer timeline here would be helpful and not require much additional text.

Line 190 – Parasitic lighting might be a photography term that I'm unaware of? If so, keep it! If not, I would say "In order to avoid interference from other light sources, photos were taken ... etc"

Line 218 – Again, I would explicitly state the timeline, either here or above (see previous comment)

Line 262 – Did any toads exude toxin during sprint trials? Was this information recorded? Speed here is considered in the context of an antipredator response where faster individuals are also bolder, however individuals that stand their ground against a predator and rely on toxin could also be considered “bold” as well. See:

Edelaar P, Serrano D, Carrete M, Blas J, Potti J, Tella JL (2012) Tonic immobility is a measure of boldness toward predators: an application of Bayesian structural equation modeling. Behav Ecol 23:619– 626

Hudson CM, Brown GP, Shine R. Evolutionary shifts in anti-predator responses of invasive cane toads (Rhinella marina). Behav Ecol Sociobiol. 2017;71: 134.

Line 341 – I would like to see some more amphibian specific locomotion studies here (e.g. speed to body size relationships). Perhaps the unique movement style of E. calamita differs from other amphibians/bufonids. Did you measure any other morphological traits (e.g. limb length) that could be relevant for speed?

Line 358 - Perhaps cite and discuss this MS:

Blennerhassett RA, Bell-Anderson K, Shine R, Brown GP. The cost of chemical defence: the impact of toxin depletion on growth and behaviour of cane toads (Rhinella marina). Proceedings of the Royal Society B: Biological Sciences. 2019;286: 20190867.

In the above study toads changed their activity level when glands were recently emptied, thus they may not be assessing their predation risk by their own gland size, but rather how recently they have deployed toxin / how much is available to them.

Line 383 - This is interesting given that other toad exploration / antipredator studies found no difference between sexes in cane toads and guttural toads, perhaps expand on this? See studies below.

Gruber J, Brown G, Whiting MJ, Shine R. Behavioural divergence during biological invasions: a study of cane toads (Rhinella marina) from contrasting environments in Hawai’i. R Soc Open Sci. 2018;5: 180197.

Baxter-Gilbert J, Riley JL, Measey J. Fortune favors the bold toad: urban-derived behavioral traits may provide advantages for invasive amphibian populations. Behav Ecol Sociobiol. 2021;75: 130.

Reviewer 3 ·

Basic reporting

I find this manuscript to be generally well written and structured, though some sentences need improved clarity of expression (listed below, I have tried to provide suggestions). There are sufficient references, and a reasonable study background has been provided to contextualize results. The manuscript figures and tables are neatly represented.

I list lines below that I would recommend improving (suggestions only):
Ln 23: change “as well” to “also”, as the former can be interpreted in multiple ways.
Ln 45: “Indeed, hardly can selection on behaviour be regarded as directional, since the fitness consequences of a given value of any behavioural trait are context-dependent.” This sentence is difficult to read, consider reversing sentence structure here for readability, or shortening the second half of the sentence.
Ln 66: change “sharks with a more exploratory behaviour” to “sharks that exhibit more exploratory behaviour”
Ln 68-71: you provide two examples here of why “the relationships between predation risk and behavioural traits may transcend plasticity,” however do not provide enough information to explain how these results don’t allow for a role for plasticity, and are different from examples above. Is it that they were done under a common garden setup? If so I would explicitly mention this.
Ln 93: “Concomitant to these substances can frequently be found aposematic colorations” sentence structure is hard to read.
Ln 94: change “associate to” to “associate with”
Ln 113: remove “One of them was locomotion”, and add to the end of the previous sentence: “with other antipredator strategies, including locomotion.”
Ln 115: You frequently use the word “moreover”, consider changing some to “furthermore” or “additionally”.
Ln 131: consider changing “… massive, and have larger…” to “… massive, or have larger…” as you test some of these things independently from one another.
Ln 144: Change “under” to “during”
Ln 158: Use of the word “formation” here is ambiguous, do you mean the forest itself, the habitat structure, or something else?
Ln 207: would recommend changing “Then, after calibrating length thanks to the piece of graph paper…” to “I calibrated length using the piece of graph paper, and manually outlined…”
Ln 357: change “did not involve” to “was to associated with”

Experimental design

The experimental design is sound, and the aims and scope appropriate for this journal. The research question is well presented both within the context of the study and regarding the background information used to direct it. However, there are some aspects of the methods that need clarification or further detail for repeatability. I have tried to provide suggestions below for improved method presentation where I can.

Additional results needing representation:
- In particular, I found the ANCOVAs (how many were run and the exact predictor/response variables in each) quite hard to follow. Please clarify this in both the Methods and Results body text. Results of the ANCOVAs should be reported in tables in the main manuscript, and this will also increase clarity.
- I would recommend including the outputs of the PCAs, including eigen values for both retained and not retained PCs, in the supplementary materials. You may also want to include a visual of the PCAs.
- I would recommend making significant p-values stand out more in all relevant graphs by e.g. making them bold.

Some further notes on improving experimental reporting:
Ln 43: “boldness or exploratory” I would be explicit here as to whether these terms are interchangeable, and if not what defines them within this manuscript.
Ln 181-198: A simple diagram figure may help convey experimental setup, and would be useful for anyone wishing to replicate these methods.
Ln 196: “calibrate longitude”, do you mean “calibrate length” as you use this term on ln 207.
Ln 212: Please be explicit here what “dorsum” refers to. Does this include legs and arms? If so, how did you ensure that limb placement didn’t impact the results?
Ln 267: How was the camera affixed for sprint speed trials (height/angle)? Please mention this in Ln 218-226. Is it at all possible that the angel the camera was recording at had an impact on estimating when the toad snout crossed the white lines (larger toads would be more affected by viewing angle if the camera was low down)?
Ln 273: “I also set a dark background at one end of the raceway, which could be viewed as a shelter and encourage toads’ moving forward” Please add a reference for this statement if one is available.
Ln 301: regarding the external squares visits/visited ratio, in line 256-257, you call this “internal squared” not “external squares”
Ln 305-308: If you have already defined the PC’s greater than 1 for both PCA 1 and 2, then refer to them by their names here (i.e. PCa, PCb). Also turn this into two sentences, and make very clear how many ANCOVAs were run and for which variables, as I found this section confusing.
Ln 330-338: You do not mention the PC for PCA2 (called PC1 I think? Table 2).
Figure 1: Include in figure legend the interpretation of PCa (what do high and low values mean).

Validity of the findings

The findings presented in this manuscript are sound, and the conclusions clear and link succinctly back up to the original experimental aims.

I would like to clarify the measure of sprint speed used as part of this study, however. I am uncertain whether the speed measurement used is the best way to represent an individual toad’s sprint speed, especially because the experimental design had no repetition on the individual level. I am cautious about using the single highest speed value as I wonder if such a small distance would be bias against smaller toads, which may require many hops to cover what a large toads would do in one. I understand size correction was performed, however there is the added factor of the “encouragement” used to promote movement. How was this done, and could this have impacted results? Was the amount of encouragement needed for each toad recorded, and could this be accounted for? Alternatively, maybe find a more robust way of measuring sprint speed, e.g. an average of the top three speeds.

Finally, it would be pertinent for the discussion to include more comparisons to other amphibian behavioural and anti-predator literature, particularly when discussing parotoid gland results.

Some brief comments on particular lines:
Ln 343: change “between speed and behavioral traits” to “between speed and the behavioral traits tested as part of this study” or something similar to acknowledge that this statement doesn’t apply to all behavioral traits.
Ln 354-356: It would be good to mention somewhere in the discussion any limitations that each of these behavioural proxy measures may carry.

---

## Round 0.2 · Minor Revisions

Overall, I think that this manuscript is looking in really good shape. I have inserted a number of – mostly writing style and structure – comments. As such, they are not mandatory, however I think they could be useful in the final polishing of this paper. When you resubmit it, you do not need to worry about a formal response/rebuttal or a tracked changes form. This round of edit is more to just provide one last round of suggestion to further increase the readability and presentation of this work.

The system will only let me upload a PDF to send you (which is not super helpful for little tracked changed edits), so I will personally email you a .doc file with tracked changes so that it makes editing on your side easier - should you agree with my suggestions.

Also, here are a few things I did not stop to edit each time, but you may want to try to adjust as you read through are:

1) The amount of time you start a sentence with a qualifier (e.g., “However”, “Yet”, “Firstly”, “Finally”, “On one hand”, and so on). I understand using qualifiers like this allow the reading to sound more conversational, which can increase readability – but if used to often the opposite effect occurs. I would suggest you work through and try to limit the amount of them here. There are some paragraphs where more than half the sentences have them. A few here and there are good. But try to use them sparsely.

2) One thing I think would help push this manuscript to the next level for readability is double checking each paragraph has a strong and encompassing topic sentence. This way the reader, at the start of each paragraph, has the ideas that are about to be discussed put into context right off the hop. This makes a world of difference. You have some great one already. But some spots could use improvement.

3) I would also suggest revising the figure and table captions, right now they are a little bare and if readers are skimming the paper, they do not provide enough detail or clarity.

---

## Round 0.3 · accepted · Accept

The paper is looking in good shape and I think we are at a point where I am happy to recommend it be accepted. I want to congratulate you on such an interesting and well put together study.

That said, after a final read I noted there is a few little typos that I wanted to flag for you to fix during proofing, there is one on line 137 ("malesare" should be "males are"), 157 ("e.g." needs a comma after it "e.g.,"), 184 (you need a space between "9" and "cm"), 244 (add spaces between "54" "x" "27" "x" "40" and "cm"), and 280 (add space between "the" and sprint"). There may well be others i did not see, so be thorough in your reading of the proofs to be sure you catch any missed one.